# Molecular Study of *Pneumocystis jirovecii* in Respiratory Samples of HIV Patients in Chile

**DOI:** 10.3390/jof10020117

**Published:** 2024-01-31

**Authors:** Isabel Iturrieta-González, Carolina Chahin, Johanna Cabrera, Carla Concha, Pamela Olivares-Ferretti, Javier Briones, Fernando Vega, Luis Bustos-Medina, Flery Fonseca-Salamanca

**Affiliations:** 1Department of Preclinic Sciences, Medicine Faculty, Laboratory of Infectology and Clinical Immunology, Center of Excellence in Translational Medicine-Scientific and Technological Nucleus (CEMT-BIOREN), Universidad de La Frontera, Temuco 4810296, Chile; isabel.iturrieta@ufrontera.cl; 2Jeffrey Modell Foundation for Diagnosis and Research in Primary Immunodeficiencies, Center of Excellence in Translational Medicine, Medicine Faculty, Universidad de La Frontera, Temuco 4810296, Chile; 3Infectology Unit, Hospital Dr. Hernán Henríquez Aravena, Temuco 4781151, Chile; johannacq2@gmail.com (J.C.); concha.carla@gmail.com (C.C.); javier.brionessegui@gmail.com (J.B.); 4Biosocial Research and Education Laboratory, Tonalli Ltd., Temuco 4810921, Chile; ps.olivares.ferretti@gmail.com; 5Critical Patient Unit, Hospital Dr. Hernán Henríquez Aravena, Temuco 4781151, Chile; fvegag@gmail.com; 6Department of Public Health and CIGES, Faculty of Medicine, Universidad de La Frontera, Temuco 4781176, Chile; luis.bustos@ufrontera.cl; 7Department of Preclinic Sciences, Medicine Faculty, Laboratory of Molecular Immunoparasitology, Center of Excellence in Translational Medicine-Scientific and Technological Nucleus (CEMT-BIOREN), Universidad de La Frontera, Temuco 4810296, Chile; flery.fonseca@ufrontera.cl

**Keywords:** *Pneumocystis jirovecii*, HIV, mutations, *DHPS*, sulfamide resistance

## Abstract

*Pneumocystis* is an opportunistic fungus that causes potentially fatal pneumonia (PCP) in immunocompromised patients. The objective of this study was to determine the prevalence of *P. jirovecii* in HIV patients through phenotypic and molecular study, to investigate the genetic polymorphisms of *P. jirovecii* at the mitochondrial gene mtLSU and at the nuclear dihydropteroate synthase gene (*DHPS*), and by analysis of molecular docking to study the effect of *DHPS* mutations on the enzymatic affinity for sulfamethoxazole. A PCP prevalence of 28.3% was detected, with mtLSU rRNA genotypes 3 (33.3%) and 2 (26.6%) being the most common. A prevalence of 6.7% (1/15) mutations in the *DHPS* gene was detected, specifically at codon 55 of the amino acid sequence of dihydropteroate synthase. Molecular docking analysis showed that the combination of mutations at 55 and 98 codons is required to significantly reduce the affinity of the enzyme for sulfamethoxazole. We observed a low rate of mutations in the *DHPS* gene, and molecular docking analysis showed that at least two mutations in the *DHPS* gene are required to significantly reduce the affinity of dihydropteroate synthase for sulfamethoxazole.

## 1. Introduction

*Pneumocystis jirovecii* (formerly *Pneumocystis carinii* f. sp. *hominis*) corresponds to an opportunistic, ubiquitous, and unicellular fungus belonging to the phylum *Ascomycota* that causes pulmonary infections in humans, including potentially fatal pneumonia (PCP) [1]. It affects immunocompromised patients, especially HIV/AIDS patients with CD4+ T-cell count < 200 cells/µL, which emerged as one of the main causes of lung infection [2]. Despite the decline in PCP incidence among HIV patients in developed countries due to the introduction of highly active antiretroviral therapy (HAART) and anti-*Pneumocystis* prophylaxis [3,4], it remains a significant pathogen. This is particularly true for undiagnosed HIV patients without HAART, as well as for those with suboptimal treatment adherence or resistance [5,6]. Extrapulmonary forms are less frequent and generally occur due to hematogenous or lymphatic dissemination of the agent, which can cause distant infections such as those in the spleen, liver, bone marrow, and paraspinal mass formation, among others [7,8,9]. Approximately 500,000 cases are described annually worldwide, and the mortality rate varies from 10–30% [10]; however, it can reach rates of 30–60% in non-HIV immunocompromised patients [11]. The signs and symptoms of PCP are non-specific and include, in most cases, fever or subfebrile temperatures, dry cough, progressive dyspnea [12,13], and bilateral ground–glass pulmonary opacities (perihilar, diffuse, and/or mosaic pattern) observed on chest computed tomography (CT) [13]. In vitro diagnosis is difficult because it is a microorganism that does not grow in culture media, and its detection is generally carried out by microscopic observation of trophic or cystic forms in respiratory samples with cytochemical or immunofluorescence staining, a method that presents a sensitivity close to 90% [10]. Serum determination of 1,3-β-D-glucan (BDG), a component of the cell wall and considered a panfungal marker, can also be used to support the diagnosis of PCP; it has high sensitivity values of 95–96%, however, its specificity is 84–86% [14]. The development of molecular tools based on conventional or nested PCR for amplification of the coding sequence of the large mitochondrial subunit mtLSU, also called mt26S of ribosomal RNA (rRNA), has considerably increased the sensitivity and specificity of its detection, reaching values of 94–100% [15,16], allowing the analysis of genetic variability through the study of genotypes and coinfections by more than one strain [17]. The introduction of Real-time PCR (RT-PCR) has substantially improved the diagnosis of PCP and has been defined as the most highly sensitive method for *P. jirovecii* detection with sensitivity and negative predictive values (NPV) of 100% [15]. Since 1990, the first-line treatment worldwide for mild, moderate, or severe PCP, as well as for its prophylaxis, is the combination of Sulfamethoxazole with Trimethoprim (SMX-TMP), which shows excellent penetration in the tissues with intravenous and oral availability, reaching comparable serum levels [18,19]. Although its extensive use as prophylaxis has decreased the incidence of PCP in HIV patients, more attention has been paid to the potential development or selection of resistant strains [20], a phenomenon that has been reflected in descriptions of refractory or resistant cases [21]. SMX inhibits the action of dihydropteroate synthase, an essential enzyme for the synthesis of folic acid, through a competitive mechanism with para-aminobenzoic acid (PABA) and TMP inhibits dihydrofolate reductase, the enzyme encoded by the *DHFR* gene which reduces dihydrofolic acid to tetrahydrofolic acid and leads to the formation of purines and eventually deoxyribonucleic acid (DNA) [22,23,24]. Various groups have focused their attention on genotypic changes in the dihydropteroate synthase (*DHPS*) gene, encoding the enzyme dihydropteroate synthase, through the study of non-synonymous mutations specifically in codons 55 and 57, which are responsible for Thr55Ala and Pro57Ser amino acid changes, respectively [25,26]. Two new mutations were recently described in India at positions 288 and 294, responsible for the amino acid changes Val96Ile and Glu98Gln, respectively [27,28]. Although the association of these mutations with the development of resistance in *P. jirovecii* is not completely clear, some studies support it [21,27,29,30]. Contrarily, there have been fewer reports of nucleotide changes in the *DHFR* gene, and there is not enough evidence to suggest that *P. jirovecii* is developing mutations related to resistance to TMP [31].

Currently, one of the most useful tools for molecular analysis of the *DHPS* gene is gene sequencing due to its high discriminatory power and reproducibility for the detection of mutations missense, nonsense, or frameshift and thus be able to monitor possible phenotypic changes in relation to the susceptibility of *P. jirovecii* against sulfonamides [26]. In Chile, there is little information about the molecular characterization of *P. jirovecii*, with most studies being epidemiological, and with a retrospective description of PCP cases [32,33]. For this reason, the present study aims to determine the prevalence of *P. jirovecii* in HIV patients with symptoms suggestive of pulmonary infection, using phenotypic and molecular methodology, to identify the different genotypes, to determine the prevalence of mutations in the gene *DHPS* and, through molecular docking analysis, its potential implication in the development of resistance to sulfonamides.

## 2. Materials and Methods

### 2.1. Studied Patients and Samples

This prospective study included adult patients diagnosed with HIV infection, from Dr. Hernán Henríquez Aravena Hospital in the city of Temuco, Chile, presenting with respiratory signs and symptoms suggestive of pulmonary infection. This study was conducted between February 2022 and August 2023 using sputum, bronchoalveolar lavage (BAL), and pharyngeal lavage samples. The samples were collected in sterile containers and transported to the Laboratory of Infectology and Clinical Immunology at the Center of Excellence in Translational Medicine of the Universidad de La Frontera for analysis.

Demographic and clinical data of the patients were collected through a survey and review of their medical records. Information regarding demographic data, such as age, sex, data related to HIV infection, CD4+ T cell count, signs and symptoms of lung infection, comorbidities, and previous use of SMZ-TMP, was obtained. 

### 2.2. Phenotypic Detection of P. jirovecii

The phenotypic detection of *P. jirovecii* in respiratory samples was carried out as outlined below: A portion of approximately 100 µL was taken from the sputum sample and deposited on a slide to make a smear with a diameter of approximately 15 mm. For both BAL and pharyngeal lavage samples, a 3 mL aliquot was transferred to an Eppendorf tube for concentration by centrifugation at 5000 rpm for 5 min. Subsequently, 100 µL of the obtained sediment was deposited onto a slide, employing the same method as for the sputum sample. The slides were then dried at room temperature and stained with Grocott-Gomori’s methenamine silver (GMS) using a Methenamine silver plating kit acc. to Gomori (Sigma-Aldrich, St. Louis, MO, USA) according to the manufacturer’s instructions. Subsequently, the microscopic examination was performed at a magnification of 1000× to search for cystic forms of *P. jirovecii*.

### 2.3. Molecular Detection of P. jirovecii

The respiratory samples were digested with proteinase K at 56 °C to extract genomic DNA using the commercial Mini kit QIAamp DNA (Qiagen, Basel, Switzerland), following the manufacturer’s instructions. The DNA obtained was stored at −20 °C until PCR amplification. As a positive control for the extraction process, human β-globin amplification was performed using primers PCO4 (5′-TCACCGCAACTTCATCCACGTTCACC-3′) and GH20 (5′-GAAGAGCCAAGGACAGGTAC-3′) [34]. Molecular identification of *P. jirovecii* was performed by PCR of the mtLSU region using specific primers pAZ102-E (5′-GATGGCTGTTTCCAAGCCCA-3′) and pAZ102-H (5′-GTGTACGTTGCAAAGTACTC-3′) [35]. Subsequently, a 1.5% agarose gel electrophoresis was performed (Kartell, Milano, Italy) stained with GelRed^®^ Nucleic Acid Gel Stain—Biotium (Fremont, CA, USA) for the visualization of a band of 346 bp indicative of the presence of *P. jirovecii* in the sample. *P. jirovecii* DNA obtained from a patient with a confirmed diagnosis of PCP through gene sequencing was used as a positive control, and the PCR mixture without the addition of DNA was used as a negative control. To confirm the presence of *P. jirovecii* in respiratory samples, the PCR product obtained was purified and sequenced using the same pair of primers utilized for amplification. Sequencing was performed at Austral-omics of the Universidad Austral de Chile (Valdivia, Chile), using the ABI Prism 310 (Applied Biosystems, Foster City, CA, USA) automated sequencer. The sequences obtained were edited using SeqMan software v. 7.0.0 (DNAStar Lasergene, Madison, WI, USA) and the consensus sequences were compared in the database of the National Center for Biotechnology Information (NCBI). An identity percentage of >99% was employed as a criterion to confirm the correct identification of fungal species.

### 2.4. Analysis of P. jirovecii Genotypes

The genotype study was conducted by analyzing mtLSU sequences of rRNA using Molecular Evolutionary Genetics Analysis (MEGA) software version 6.0. [36]. The sequences were aligned using the Clustal W algorithm [37] and refined with MUSCLE in the same platform [38]. The analysis was performed based on nucleotide polymorphisms at two informative positions in the sequence (85 and 248). Thus, the presence of genotype 1 (C/C), genotype 2 (A/C), genotype 3 (T/C), and genotype 4 (C/T) were studied [39]. Phylogenetic reconstruction was performed using Maximum Likelihood (ML) using MEGA software. The best nucleotide substitution model was determined on the same platform, and mtLSU sequences from *Pneumocystis murina* species (HLSU6 and HLSU7) were used as an outgroup reference. These analyses allowed additional determination of the presence of mixed infections. The mtLSU sequence data generated in this study were deposited in GenBank.

### 2.5. Study of Mutations in the DHPS Gene

Samples positive for *P. jirovecii* were amplified for the *DHPS* gene using the primers DHPS-3 (5′-GCGCCTACACATATTATGGCCATTTTAAATC-3′) and DHPS-4 (5′-GGAACTTTCAACTTGGCAACCAC-3′) [40]. The amplified fragment equivalent to 370 bp was visualized using 1.5% agarose gel electrophoresis (Kartell, Italy), stained with GelRed^®^ Nucleic Acid Gel Stain—Biotium (Fremont, CA, USA), and sequenced using the same pair of primers utilized in the amplification. Subsequently, the sequences were edited to obtain the consensus sequence and aligned using Molecular Evolutionary Genetics Analysis software (MEGA) version 6.0 [36], as mentioned in Section 2.4. For mutational analysis of the *DHPS* gene, wild type reference sequences (AY628435) and sequences with previously described mutations in codons 55 and 57 (U66281), as well as codons 96 and 98 (MG010799), were included to study the presence of wild type strains, single mutants or with several mutations. The *DHPS* gene sequence data generated in this study were deposited in the GenBank.

### 2.6. In Silico Analysis of Dihydropteroate Synthase Enzyme 

To investigate the impact of *DHPS* mutations, the amino acid sequence was obtained using the Expasy tool (https://web.expasy.org/translate/), accessed on 9 July 2023, with AAF14263 as the reference sequence from the NCBI database (www.ncbi.nlm.nih.gov/protein/), accessed on 9 July 2023. The three-dimensional structures of both the wild type (as the positive control) and mutants were generated using the AlphaFold2 method [41], implemented in ChimeraX v1.4 (https://www.rbvi.ucsf.edu/chimerax), accessed on 12 July 2023 and executed on Google Colab servers [42]. The resulting dihydropteroate synthase models were refined using Swiss-PDBViewer software v. 4.1.0 [43].

For molecular docking analysis, the binding site of the enzyme was determined employing the Protein–Ligand Interaction Profiler [44] based on the interaction between the ligand ID: 78H and the enzyme model ID: 3TYA retrieved from the Protein Database (https://www.rcsb.org/) accessed on 19 July 2023. The docking box was defined using ADFRsuite v1.0, with dimensions of 25 × 25 × 25 Å assigned to each model. The ligands utilized were sulfamethoxazole (CID: 5329), 7,8-dihydropterin (dihydropterin, CID: 135440520), 4-aminobenzoic acid (PABA, CID: 978), and dimethyl sulfoxide (DMSO, CID: 679) downloaded from PubChem [45] and optimized and prepared using OpenBabel [46] employing the mmff94 force field [47]. Molecular docking calculations were performed using AutoDock Vina v. 1.2.3 [48], with an exhaustiveness value set to 32. 

To assess the affinity of all ligands for each enzyme model, statistical analysis was conducted using GraphPad PRISM 7.0 software (GraphPad Software, La Jolla, CA, USA), encompassing one-way ANOVA, Dunnett’s multiple comparisons test, with a significance criterion of *p* < 0.05. PyMOL (The PyMOL Molecular Graphics System, Version 2.3.1 Schrödinger, LLC, New York city, NY, USA) was used for visualization.

### 2.7. Statistical Analysis

Statistical analysis was performed using Stata software version 18.0. Qualitative variables were described based on absolute frequencies and proportions. Continuous variables were described providing the range, mean, and standard deviation. Differences between continuous variables in patients with and without a diagnosis of *P. jirovecii* infection were determined using Student’s *t*-test. The proportions of these groups were compared using the χ^2^ test. Differences were considered statistically significant at *p* < 0.05.

### 2.8. Ethical Aspects

This study was approved by the ethics committee of Servicio de Salud Araucanía Sur (protocol no. 260, approved on 9 November 2021). All patients included in the study provided their approval to participate by signing an informed consent form.

## 3. Results

### 3.1. Patients Studied, Clinical and Demographic Information

A total of 53 adult patients diagnosed with HIV and lower respiratory tract infection were included in this study with ages between 23 and 66 years. The Analysis included 53 respiratory samples, 38 of which were sputum, 13 BAL, and 2 pharyngeal lavage samples. There was no statistically significant association between the sex of the patient and the development of *P. jirovecii* infection. Most of the patients (43.4%, *n* = 23) were between 40 and 49 years old; however, no association was observed between the patient age group and *P. jirovecii* infection (*p* = 0.904). Within the associated comorbidities, candidiasis, alcoholism, and severe malnutrition or wasting syndrome were the most frequent, however, no significant difference was observed between the comorbidities presented by patients with and without *P. jirovecii* infection. 73.6% of the patients (39/53) presented CD4+ T-cell count < 200 cells/uL. The clinical and demographic data of the patients included in this study are summarized in Table 1.

### 3.2. Characterization of Patients with Positive Detection of P. jirovecii in Respiratory Samples

*P. jirovecii* was detected in 28.3% of the samples studied (15/53). In 13 patients, *P. jirovecii* was detected by histological demonstration of cystic forms of the agent in samples stained with GMS and through PCR amplification of the mtLSU region of the rRNA. In the two remaining patients, the diagnosis was achieved only through PCR on the sputum sample and pharyngeal lavage (Figure 1).

Most of the patients positive for *P. jirovecii* (53.3%, *n* = 8), were between 40 and 49 years of age and were mainly male (93.3%, *n* = 14). It was observed that 46.7% of HIV patients did not present with other associated comorbidities. With a single exception, all cases demonstrated a CD4+ T cell count < 200 cells/µL, with an average of 40 ± 52.7 cells/uL. Among the respiratory symptoms presented by patients in whom *P. jirovecii* was detected, the most frequent were cough with or without expectoration (73.3%, *n* = 11) and dyspnea (66.7%, *n* = 10). The most frequently observed radiological finding on CT was bilateral ground–glass opacity (Figure 2). However, one patient presented multiple solid nodules with random and perilymphatic distribution. In addition, 12 of the 15 HIV-seropositive patients positive for *P. jirovecii* had not received HAART before the diagnosis of PCP. The treatment of the 15 patients with *P. jirovecii* predominantly involved oral or intravenous administration of SMX-TMP over a span of up to 21 days. Instances of adverse drug reaction (ADR) and renal failure prompted a shift to treatment involving primaquine (15–30 mg/day) in combination with clindamycin (600 mg) for 21 days. Additionally, due to the clinical deterioration observed in two patients, an echinocandin (caspofungin or anidulafungin) was administered together with SMX-TMP. Regarding patient management, 13 (86.7%) patients required hospitalization and were admitted to the Medicine service (7 patients) or to the critical patient unit (CPU) (6 patients). Most patients with PCP presented a favorable evolution (12/15); however, three individuals died due to respiratory failure, yielding a mortality rate of 20% (refer to Table 2).

### 3.3. P. jirovecii Genotypes Based on rRNA mtLSU

Analysis of the genotypic variants of *P. jirovecii* was performed based on the polymorphisms observed at nucleotides 85 and 248 of the sequences of the mtLSU region of the rRNA and through phylogenetic analysis. For phylogenetic construction, the best nucleotide substitution model determined using the MEGA software was the Tamura 3-parameter with gamma distribution (T92 + G). The analyses allowed us to identify the presence of 4 genotypes, with genotype 3 being the most frequent, detected in 33.3% (5/15) of the patients, followed by genotypes 2 and 1 detected in 26.6% (4/15) and 20% (3/15) of the patients, respectively (Table 3). The least frequent genotype was 4, which was detected in two patients, and a mixed infection with the presence of genotypes 1 and 2 was detected (Figure 3).

### 3.4. Study of Mutations in the DHPS Gene

*DHPS* was amplified and sequenced from 15 samples positive for *P. jirovecii*. Genetic analysis of the sequence made it possible to show that 14 of them did not present mutations (93.3%), corresponding to wild type strains (Table 4). The presence of a missense mutation in codon 55 (Thr55Ala) was identified in one of the samples studied, from a patient with bilateral pneumonia associated with COVID-19, hospitalized in the ICU and connected to mechanical ventilation, and who did not show a good evolution, resulting in fatality (case 8, Table 2).

### 3.5. In Silico Analysis Results

Using AlphaFold2, enzymatic models of wild type dihydropteroate synthase and 15 mutant models were predicted with the mutations Thr55Ala, Pro57Ser, Val96Ile, and Glu98Gln. Amino acid change mutations are highlighted in red (Figure 4A). The Thr55Ala and Pro57Ser mutations are found in loop 2, both close to the active site of the enzyme, whereas Val96Ile and Glu98Gln are in the α3 helix, located outside the β-strand pocket.

Additionally, as part of our control experiments, we conducted docking analyses involving the wild type enzyme model with sulfamethoxazole, dihydropterin, PABA, and DMSO. The Appendix A reveals significant differences (*p* < 0.05) in binding affinities, with sulfamethoxazole exhibiting higher affinity compared to the other molecules. Particularly, dihydropterin and PABA are substrates of dihydropteroate synthase, while DMSO serves as a negative control, which lacks antifungal activity on its own. The observed binding affinity of sulfamethoxazole in these experiments provides a relevant benchmark for evaluating the binding affinities of sulfamethoxazole with the mutant dihydropteroate synthase models.

The results of the docking analysis of the enzymatic models of dihydropteroate synthase facing the sulfamethoxazole complex showed that the presence of a single mutation of Thr55Ala, Pro57Ser, Val96Ile or Glu98Gln does not significantly affect the binding affinity to sulfamethoxazole. Only the Thr55Ala-Glu98Gln model showed a lower binding affinity compared to the wild type enzyme, with energy values of −6.31 kcal/mol and −6.67 kcal/mol, respectively (Figure 4B).

Nucleotide sequences obtained in the present study were submitted to NCBI GenBank. The accession numbers are mtLSU: OR387295 to OR387308; OR499861; OR478965. *DHPS*: OR398332 to OR398344; OR487488; OR501063.

## 4. Discussion

*P. jirovecii* pneumonia is a common opportunistic fungal infection in immunocompromised patients. Its prevalence in patients with HIV has undergone significant changes in recent years, with a reduction in its incidence mainly due to the introduction of antiretroviral therapy in 1996 and prophylactic therapy with SMZ-TMP in 1989 [49,50]. It is important to consider that in underdeveloped or developing countries in which access to HIV diagnosis and treatment is limited, *P. jirovecii* continues to be an important cause of lung infection with potential evolution to respiratory failure and death [51,52].

In the present study, an infection prevalence rate of 28.3% by *P. jirovecii* was documented among HIV patients, similar to that previously described in Brazil, where a prevalence of 23.5% and 26.3% was reported between 2009 and 2014, respectively, in patients from various hospitals from Rio de Janeiro [53,54]. We also observed that approximately 50% of the positive cases detected in our series corresponded to new HIV diagnoses. Therefore, in our environment, PCP appears as one of the main opportunistic diseases that defines the AIDS stage. There was no difference in the average age of patients with and without *P. jirovecii* infection; however, in relation to the CD4+ T lymphocyte count, we observed that although the average CD4+ T lymphocyte count was <200 cells/µL, patients with lung infection by *P. jirovecii* showed an average value of <50 cells/µL (40 cells/µL), with a statistically significant difference (*p* = 0.0036). 

We observed a robust correlation between the phenotypic diagnostic technique based on microscopic observation of cystic forms of *P. jirovecii* and the molecular approach, relying on conventional PCR amplification of the mtLSU region with only two cases identified solely through PCR (Cases 7 and 15) (Table 2), like what was recently described by [55]. The mitochondrial genome of *P. jirovecii* replicates and transcribes for the most part autonomously and encodes mainly proteins involved in mitochondrial respiration, its high conservation facilitates not only agent identification but also genotype exploration. Differences in genotype distribution have been noted based on geographical location. The most prevalent mtLSU genotype in isolates from Santiago in Chile was genotype 4 [33]. In our investigation performed in the south of Chile, genotypes 2 and 3, although with a slightly higher frequency than genotype 1, emerged as the most frequent within our HIV patient cohort, similar to what has been described in countries such as Italy [56,57], Tunisia [58] or India [59]. This distribution of genotypes differs from that reported in Korea or France where genotype 1 predominates [60,61]. Mixed infections by *P. jirovecii*, remain infrequent [40,56,59], a trend also reflected in our study, where only one case exhibited genotype 1 and 2 coinfection. The clinical impact of these genetic variants is not completely clear; however, it has been observed that in colonized children and HIV/AIDS adults, there are significant differences in terms of the identified genotype, with a predominance of 3 (42.1%) and 2 (42.3%), respectively [62]. Similarly, a clear predominance of genotype 1 has been reported in transplant recipients in India [59]. Future studies are needed to accurately understand and clarify its association with the development of infection in a certain group of patients or potential differences in the expression of virulence or antimicrobial resistance.

Susceptibility studies on *P. jirovecii* are limited because of their inability to grow in standard In vitro cultures. Consequently, various studies carried out worldwide based on molecular analysis have been developed to detect possible genotypic changes that may generate variations in their sensitivity patterns. Mutations especially those which occur in the binding site of enzymes involved in the metabolization of certain drugs, play an important role in the drug’s efficacy due to often decreasing the binding affinity between the protein and the antimicrobial molecule and therefore may be related to the development of resistance [63,64,65]. In this sense, several studies have focused on mutations in the *DHPS* gene, which encodes the enzyme dihydropteroate synthase, which has been described for its potential association with the development of resistance to sulfonamides and is currently considered the drug of choice for the prophylaxis and treatment of PCP [66]. This gene has been the main focus of study since it has been observed in animal models that most of the anti-*Pneumocystis* effects are due to the action of sulfamethoxazole, which inhibits the enzyme dihydropteroate synthase. Mutations in the *DHPS* gene have been shown to confer resistance to sulfa drugs in other microorganisms, such as *Plasmodium falciparum, Escherichia coli*, and *Streptococcus pneumoniae* [67,68,69]. The prevalence of *DHPS* mutations varies geographically. Various studies conducted in Asian countries have reported low mutation rates. A study conducted in Korea between 2007–2013 on 129 patients diagnosed with PCP described the presence of wild type strains in 100% of cases [60]. A recent Chinese study of 60 HIV patients with *P. jirovecii* infection, identified a single case with a mutation in codon 55 of *DHPS* [70]. In Africa, although similar results have been described with a prevalence of only 7.1% (1/14) in countries such as Zimbabwe [71], in 2012, 100% (13/13) of strains with mutations in the *DHPS* gene were reported in Uganda [72]. In Europe, a prevalence of 2.8% was described in Germany [73], 4.5% in Belgium [74], and 20.4% in Denmark [75]. On the American continent, the prevalence is variable, with the USA reporting the highest rates [26]. In Chile, the only existing report was developed in 2017 by Ponce et al. [32] and included a total of 46 HIV patients from the city of Santiago (Metropolitan Region), describing a prevalence of mutations of 48%, corresponding mainly to double mutants (55 and 57) and coinfections of mutant and wild type strains, presumably acquired as a result of interhuman transmission because these patients had no history of prior therapy with SMX-TMP [32]. Our results showed that there are notable differences depending on the geographical region, our study was conducted in a city in the south of Chile (La Araucanía region) and describes a prevalence of mutations in the *DHPS* gene of only 6.7%, with a single amino acid change responsible for the substitution of threonine by alanine in codon 55 of the protein sequence. Although it has not yet been possible to establish a direct association between the presence of mutations in the *DHPS* gene and the development of resistance to sulfonamides in *P. jirovecii*, it has been possible to demonstrate a correlation with greater severity of PCP, worse prognosis [76] and lower 3-month survival rates [75].

Worldwide, 4 missense type mutations (Thr55Ala, Pro57Ser, Val96Ile, and Glu98Gln) have been described in the *DHPS* gene, located in a highly conserved region of the enzyme [26,27,28]. The information regarding the clinical relevance of *P. jirovecii* remains controversial. However, an In vitro study by Moukhlis et al. [77], who used a mutant strain of *Saccharomyces cerevisiae* as a model, containing mutations 55 and 57 separately and in combination, demonstrated that a decrease in susceptibility is only evident in *S. cerevisiae* when both mutations are present simultaneously. Recently Singh et al. [27] described the association of mutant 98 (genotype G to C) with severe PCP. Our in silico study through molecular docking analysis showed that a single mutation of the four positions described (codons 55, 57, 96, or 98) is not capable of generating a significant reduction in affinity for sulfamethoxazole and therefore, on their own are unlikely to be responsible for resistance development. However, the combination of Thr55Ala-Glu98Gln significantly reduces the affinity of the enzyme for sulfamethoxazole. Although our in silico analysis was carried out including different models of the dihydropteroate synthase from *P. jirovecii* including isolated mutations as well as different combinations, further studies are necessary to analyze the correlation between *DHPS* genotype and clinical treatment response to elucidate the real behavior of this opportunistic pathogen in the presence of these amino acid changes.

## 5. Conclusions

The present study describes the prevalence of pulmonary infection by *P. jirovecii* in 28.3% of HIV patients with a low rate of mutations in the *DHPS* gene. The in silico study through molecular docking showed that a single point mutation in codons 55, 57, 96, or 98 of the *DHPS* gene did not have a significant impact on the binding affinity to sulfamethoxazole, however, the combination of mutations Thr55Ala-Glu98Gln significantly reduces the affinity for the drug. It is necessary to carry out future studies to better monitor and comprehend the behavior of *P. jirovecii*. Therefore, it is possible to detect changes in its genome in a timely manner, with a potential association with the development of resistance to antimicrobial drugs.

## Figures and Tables

**Figure 1 jof-10-00117-f001:**
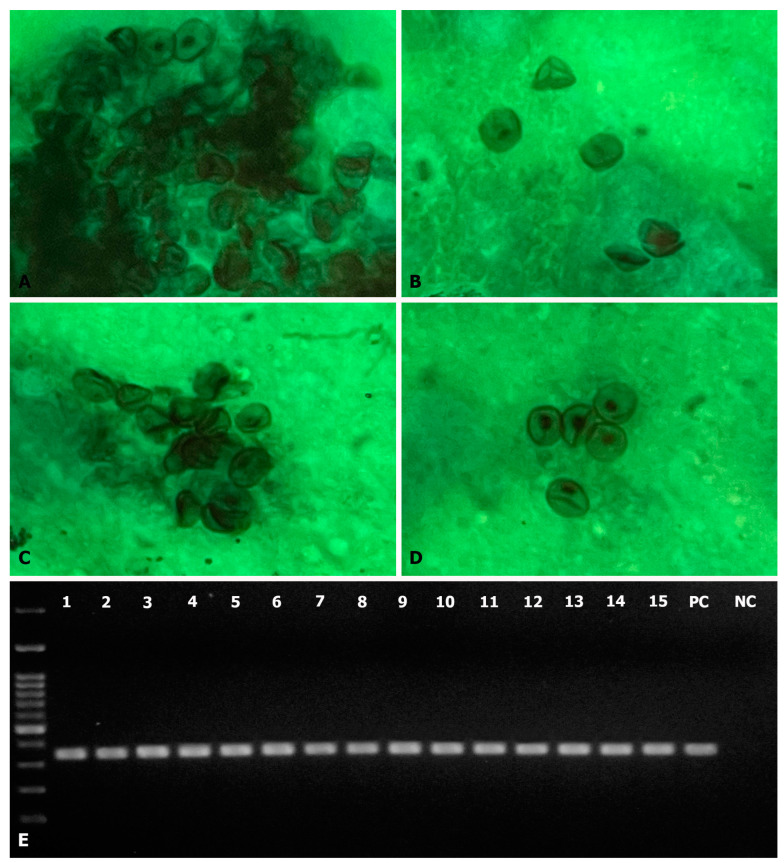
Phenotypic and molecular identification of *P. jirovecii*. (**A**–**D**) Representative images of the studied respiratory samples stained with GMS in which abundant cystic forms of *P. jirovecii* were observed (magnification × 1000). (**E**) Molecular detection via PCR amplification of the mtLSU region of the rRNA (100 bp molecular weight marker). A band of 346 bp indicated the presence of *P. jirovecii*. Lanes 1–15: positive samples; PC: positive control; NC: negative control.

**Figure 2 jof-10-00117-f002:**
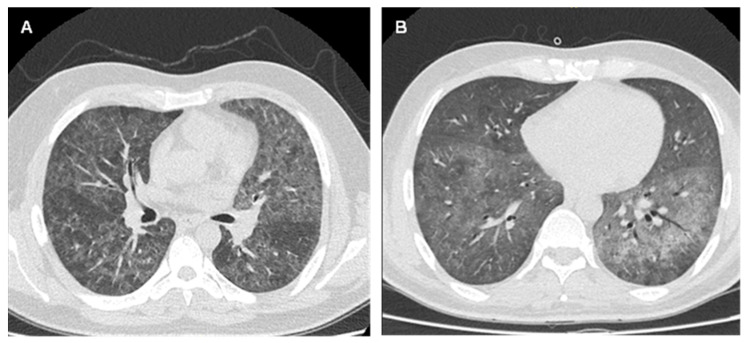
Chest CT with representative images of patients with lung infection by *P. jirovecii* obtained at the time of PCP diagnosis. (**A**,**B**) Presence of bilateral pulmonary ground glass pattern observed in patients No. 2 and No. 3 of the Table 2, respectively.

**Figure 3 jof-10-00117-f003:**
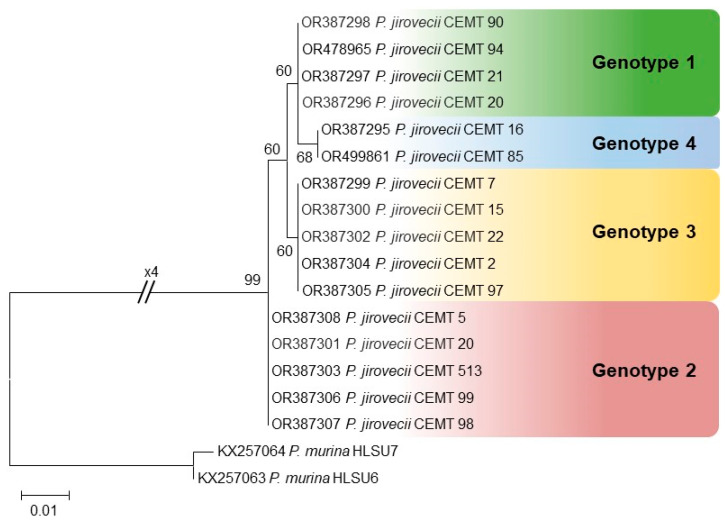
Phylogenetic tree including sequences of the mtLSU region of *P. jirovecii* strains identified from respiratory samples of HIV patients. Bootstrap values above 60% are shown at the nodes. *Pneumocystis murina* mtLSU sequences were used as outgroup.

**Figure 4 jof-10-00117-f004:**
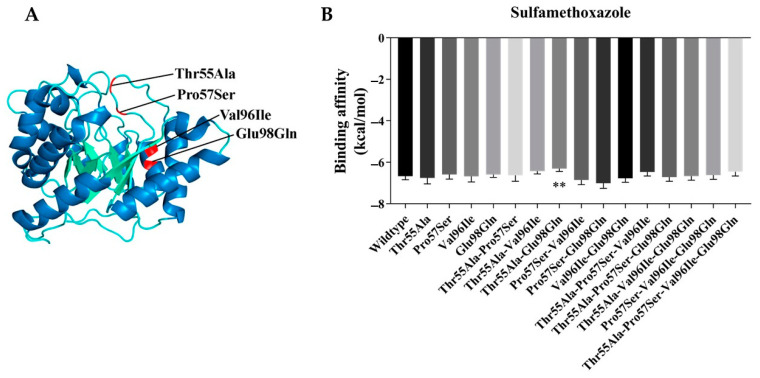
(**A**) Dihydropteroate synthase enzyme model obtained through in silico analysis, highlighting amino acid mutations in red. (**B**) Binding affinities resulting from molecular docking of dihydropteroate synthase and its mutant against sulfamethoxazole. Significant differences are indicated by asterisks (**), which denote the level of significance (*p* < 0.05).

**Table 1 jof-10-00117-t001:** Clinical and demographic data of the 53 patients with HIV included in the study.

Variable	Total *n* (%)	PCP− *n* (%)	PCP+ *n* (%)	*p* Value
No. of patients	53 (100%)	38 (71.7%)	15 (28.3%)	
Age (years)				0.904
<30	9 (16.9%)	7 (77.8%)	2 (22.2%)	
30–39	6 (11.3%)	4 (66.6%)	2 (33.3%)	
40–49	23 (43.4%)	15 (65.2%)	8 (34.8%)	
50–59	11 (20.8%)	9 (81.8%)	2 (18.2%)	
≥60	4 (7.6%)	3 (75%)	1 (25%)	
Mean ± SD	43.3 ± 11.1	43.4 ± 11.4	43 ± 10.8	0.902
Sex				0.149
Female	11 (20.8%)	10 (90.9%)	1 (9.1%)	
Male	42 (79.2%)	28 (66.6%)	14 (33.3%)	
Comorbidities or associated conditions				
Candidiasis	6	4	2	1.000
Ethylism	6	5	1	0.662
Severe malnutrition or wasting syndrome	5	3	2	0.614
Treated tuberculosis	3	2	1	1.000
Drug addiction	2	-	2	0.076
COVID-19	2	-	2	0.076
Thyroid pathology	2	2	-	1.000
Arterial hypertension	1	1	-	1.000
CNS infection	2	2	-	1.000
Hematologic neoplasm	2	2	-	1.000
Pulmonary aspergillosis	1	1	-	1.000
CMV infection	4	1	3	0.064
Treated syphilis	1	1	-	1.000
Tuberculosis	1	1	-	1.000
Smoking	1	1	-	1.000
COPD	1	1	-	1.000
Neither	23	16	7	0.769
CD4+ T-cell count (cells/uL)				
Range	0–643	0–643	4–206	
Average ± SD	118.3 ± 165.2	146.4 ± 181.9	40 ± 52.7	0.0036 *

CMV: Cytomegalovirus; CNS: central nervous system; COPD: Chronic obstructive pulmonary disease; SD: standard deviation; *: variable with significant difference with *p*-value < 0.05.

**Table 2 jof-10-00117-t002:** Clinical and demographic characteristics of HIV patients in whom *P. jirovecii* was detected in respiratory samples.

Patient (Code)	Age/Sex	Sample	Genotype mtLSU	Period since HIV Diagnosis	CD4+ T Cell (cell/uL)	Respiratory Symptoms/Clinical Signs	Previous Use SMX-TMP	Patient Management	Treatment	Evolution
1 (CEMT21)	40/M	BAL	1	New diagnosis	206	CT with GGO.	No	Hospitalization (Medicine)	SMX-TMP 800/160, 1 tablet every 8 h orally for 7 days. Due to renal failure, change to PQ 15 mg/day + CLDM 600 1 tablet every 8 h. Total duration 21 days	Survived
2 (CEMT20)	34/M	Sputum	1 and 2	8 years	31	Cough with expectoration, dyspnea. CT with bilateral GGO.	W.I	Hospitalization (Medicine)	SMX-TMP 800/160, 2 tablets every 8 h orally. Due to severe ADR, switch to PQ 30 mg/day + CLDM 600 mg 1 tablet every 8 h. Total duration 21 days.	Survived
3 (CEMT16)	45/M	Sputum	4	New diagnosis	10	Progressive irritating cough, dyspnea, feeling feverish. CT with bilateral GGO.	Yes	Hospitalization (CPU)	SMX-TMP 400/80, 3 vials every 12 h IV, associated with CAS and CLDM due to clinical deterioration. Total duration 21 days	†
4 (CEMT15)	30/M	Sputum	3	11 years	4	Dry cough, exertional dyspnea for 1 month, night sweats. CT with interstitial infiltrate and bilateral GGO	No	Hospitalization (Medicine)	SMX-TMP 400/80, 4 vials every 8 h IV. Total duration 21 days	Survived
5 (CEMT7)	61/M	Sputum	3	New diagnosis	W.I	Respiratory failure and interstitial pneumonia	No	Hospitalization (CPU)	SMX-TMP 400/80, 10 vials per day + prednisone 40 mg per day. Duration 4 days (patient dies)	†
6 (CEMT22)	23/M	BAL	3	5 months	21	Cough, dyspnea, night fever. CT with condensing image, scarce GGO.	No	Hospitalization (Medicine)	SMX-TMP 400/80, 4 vials every 8 h IV for 11 days. Change to PQ 30 mg/day + CLDM 600 mg, 1 tablet every 8 h. Total duration 21 days.	Survived
7 (CEMT2)	57/M	Sputum	3	11 years	38	Cough with expectoration, dyspnea, feeling feverish. CT with bilateral GGO, mostly in the left lung	No	Ambulatory	SMX-TMP 800/160, 2 tablets every 8 h orally. Total duration 21 days.	Survived
8 (CEMT513)	44/M	BAL	2	New diagnosis	9	Bilateral pneumonia. Chest X-ray with bilateral infiltrates with nodular condensing images	No	Hospitalization (CPU)	SMX-TMP 400/80, 4 vials every 6 h IV. Duration 15 days (pacient dies).	†
9 (CEMT5)	49/M	BAL	2	1 month	29	Cough with expectoration, dyspnea, and fever	No	Hospitalization (CPU)	SMX-TMP 15 mg/kg/day IV for 18 days and then 3 days orally at the same dose. Hydrocortisone 100 mg IV every 8 h for 5 days, then 50 mg every 8 h for 5 days, and then prednisone 20 mg/day orally.	Survived
10 (CEMT99)	41/M	Sputum	2	New diagnosis	45	Cough, tachypnoea, CT with ground glass infiltrate	No	Hospitalization (CPU)	SMX-TMP 800/160 in doses of 17 mg/kg in 3 doses (of the TMP component). Total duration 21 days.	Survived
11 (CEMT98)	28/M	BAL	2	New diagnosis	9	Pneumonia with cough, dyspnea, fever, and night sweats. Chest X-ray shows multiple solid nodules of random and perilymphatic distribution	No	Hospitalization (Medicine)	SMX-TMP 800/160, 2 tablets every 8 h orally to complete 17 mg/kg of TMP. Total duration 21 days	Survived
12 (CEMT97)	46/F	BAL	3	12 years	54	Multifocal pneumonia, respiratory failure. CT chest with bilateral GGO	No	Hospitalization (CPU)	SMX-TMP 400/80, 3 vials every 6 h IV. ANI was associated. Due to clinical deterioration, CAS and CLDM were added. Total duration 21 days.	Survived
13 (CEMT90)	43/M	Sputum	1	2 years	12	Cough, dyspnea, sore throat, diaphoresis, and desaturation. CT shows diffuse bilateral GGO	No	Hospitalization (Medicine)	SMX-TMP 400/80, 3 vials every 8 h IV for 12 days. Then 800/160 2 tablets every 8 h orally. Due to toxicity, pancytopenia, and transaminase elevation, change to PQ 15 mg/day + CLDM 600 mg 1 tablet every 8 h. Total duration 21 days.	Survived
14 (CEMT85)	49/M	Sputum	4	New diagnosis	W.I	Cough with expectoration, dyspnea at rest, feeling feverish. Chest CT reveals extensive areas with bilateral GGO	No	Hospitalization (Medicine)	SMX-TMP 400/80, IV and then 800/160 2 tablets every 8 h orally. Total duration 21 days.	Survived
15 (CEMT94)	55/M	Pharyngeal lavage	1	4 years	52	Dry cough, dyspnea, tiredness, loss of appetite. Chest X-ray with interstitial infiltrates	Yes	Ambulatory	SMX-TMP 800/160: 2-2-1 (tablets) every 8 h orally. Total duration 21 days.	Survived

ADR: adverse drug reaction; ANI: Anidulafungin; CAS: Caspofungin; CEMT: Centro de Excelencia en Medicina Traslacional; CLDM; Clindamycin; CPU: Critical Patient Unit; CT: computed tomography; F: Female; GGO: Ground–glass opacity; HIV: human immunodeficiency virus; IV: intravenous; M: Male; PQ: Primaquine; SMX-TMP: sulfamethoxazole trimethoprim; W.I: Without Information; †: Fatal outcome.

**Table 3 jof-10-00117-t003:** *P. jirovecii* genotypes based on the analysis of the mtLSU region of rRNA detected in respiratory samples of HIV patients from Temuco (Chile).

mtLSU rRNA Genotype	Nucleotide and Position	Patients *n* (%)
1	85/C; 248/C	3 (20%)
2	85/A; 248/C	4 (26.6%)
3	85/T; 248/C	5 (33.3%)
4	85/C; 248/T	2 (13.3%)
Mixed ^a^	85/C; 248/C 85/A; 248/C	1 (6.7%)

A: Adenine; C: Cytokine; T: thymine; ^a^: Patient with genotype 1 and 2 infection of *P. jirovecii*.

**Table 4 jof-10-00117-t004:** *P. jirovecii* genotypes based on analysis of the *DHPS* gene detected in respiratory samples of HIV patients from Temuco (Chile).

Genotype *DHPS*	Nucleotide Position/Amino Acid Position				Patients *n* (%)
	165/55	171/57	288/96	294/98	
Wild type	A (Thr)	C (Pro)	G(Val)	G(Glu)	14 (93.3%)
Mutant (55)	**G(Ala)**	C (Pro)	G(Val)	G(Glu)	1 (6.7%)
Mutant (57)	A (Thr)	**T (Ser)**	G(Val)	G(Glu)	-
Mutant (96)	A (Thr)	C (Pro)	**A(Ile)**	G(Glu)	-
Mutant (98)	A (Thr)	C (Pro)	G(Val)	**C(Gln)**	-

Ala: Alanine; Gln: Glutamine; Glu: Glutamate; Ile: Isoleucine; Pro: Proline; Ser: Serine; Thr: Threonine; Val: Valine. Mutations found in each genotype are highlighted in bold indicating the nucleotide and amino acid change.

## Data Availability

Data are available in the Laboratory of Infectology and Clinical Immunology of the Center of Excellence in Translational Medicine (CEMT), Universidad de La Frontera, Temuco, Chile.

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
