# Peer review of "Molecular Study of Pneumocystis jirovecii in Respiratory Samples of HIV Patients in Chile"

_jof, 2024, doi:10.3390/jof10020117_

Round 1
Reviewer 1 Report
This is an interesting and valuable study that contributes information on the prevalence of Pneumocystis in HIV-related pneumonia in a city located 800 kilometers South of Santiago and, on the molecular characteristics of the isolates. The authors collected 53 clinical specimens from adults with HIV and symptoms of lower respiratory tract infection.
The main clinical findings of the study are:
1. 28% of PCP in HIV-related pneumonia in Temuco City located 800 km South of Santiago. : 15 (28.3%) of 53 patients.
2. Describes 4 types of rRNA mtLSU genotypes of P. jirovecii in their patients: Genotype 3 (in 5 patients); Genotype 2 (in 4); Genotype 1 (in 3); and Genotype 4 (in 2) were detected as unmixed. Mixed genotypes were detected in 1 patient.
3. 6% prevalence of DHPS mutations: A single mutation was found in 1(6.7%)of 15 isolates. This isolate was recovered from a patient who did not receive prior sulfa prophylaxis (Pt #8 deceased).
4. The co-existence of 2 DHPS mutations consisting of changes in amino acids reduced the building affinity of sulfa drugs to the DHPS binding site. This application of an in vitro modeling to study the different mutations in an appealing AlphaFold2 in-silico model.
A few issues that we hope will improve the manuscript before publication:
Abstract:
Given that the natural history of Pneumocystis infection in humans documents that this fungus affects many hosts, the first sentence of the abstract may read better as: "Pneumocystis is an opportunistic fungus that causes potentially fatal pneumonia (PCP) in immunocompromised patients". The objective of ....
Please review the reference by Dr. Pasic L et al. in J Fungi 2020(6):259 (doi:10.3390/jof6040259) and remove the sentence "This is the first report of P. jirovecii genotypes in patients with HIV in Chile".
Lines 423-424 should be also deleted.
Stating that Genotypes 2 and 3 are the most prevalent in Temuco and are different from previous reports from other geographical locations shall be commented on.
Manuscript:
Please quote ref #72 and Dr. Pasic in line 90. (after PCP cases).
Line 65: Reference 17 does not refer to nested PCR. Please remove nested PCR or change the reference to describe the molecular tools available to characterize Pneumocystis jirovecii isolates from immunocompromised patients. Another paper from Dr. Beard is suggested: "Strain Typing Methods and Molecular Epidemiology of Pneumocystis Pneumonia" in EID 2004.
When describing the history of TMP/SMZ as a treatment of PCP please quote the original manuscript by Hughes WT in Can Med Assoc Journal 1975.
Line 205
Reads: ...patients diagnosed with HIV were included....
Should read: ... diagnosed with HIV and lower respiratory tract infection were included....
Were the patients consecutive?
Is the etiology of respiratory infection in non-PCP cases available? Was the etiology determined in a proportion of the cases? Which etiologies?
Discussion:
Line 320: Reference 49 refers to antiretroviral therapy. Please add a reference to indicate that prophylactic therapy with SMZ-TMP started in 1989. The description of TMP SMZ prophylaxis was published in 1975.
Line 343 - 346 Single-round PCR has an excellent correlation with microscopy in HIV-infected patients please comment on this and provide a reference.
Line 352 The most prevalent mtLSU genotype in isolates from Santiago in Chile was genotype 4. (line 352) (Dr. Pasic et. al).
The manuscript would benefit from a review of English grammar. For example:
Line 40: Reads: ..., which has emerged as one of the main... "has" should be deleted as this happened in the 1980s or 1990s.
line 46 Reads: dis-semination... The hyphen should be deleted. Same as in line 163.
Author Response
Reviewer 1.
This is an interesting and valuable study that contributes information on the prevalence of Pneumocystis in HIV-related pneumonia in a city located 800 kilometers South of Santiago and, on the molecular characteristics of the isolates. The authors collected 53 clinical specimens from adults with HIV and symptoms of lower respiratory tract infection.
The main clinical findings of the study are:
- 28% of PCP in HIV-related pneumonia in Temuco City located 800 km South of Santiago. : 15 (28.3%) of 53 patients.
- Describes 4 types of rRNA mtLSU genotypes of P. jirovecii in their patients: Genotype 3 (in 5 patients); Genotype 2 (in 4); Genotype 1 (in 3); and Genotype 4 (in 2) were detected as unmixed. Mixed genotypes were detected in 1 patient.
- 6% prevalence of DHPS mutations: A single mutation was found in 1(6.7%)of 15 isolates. This isolate was recovered from a patient who did not receive prior sulfa prophylaxis (Pt #8 deceased).
- The co-existence of 2 DHPS mutations consisting of changes in amino acids reduced the building affinity of sulfa drugs to the DHPS binding site. This application of an in vitro modeling to study the different mutations in an appealing AlphaFold2 in-silico model.
A few issues that we hope will improve the manuscript before publication:
Abstract:
Given that the natural history of Pneumocystis infection in humans documents that this fungus affects many hosts, the first sentence of the abstract may read better as:
Comment 1: "Pneumocystis is an opportunistic fungus that causes potentially fatal pneumonia (PCP) in immunocompromised patients". The objective of ....
According with the reviewer´s recommendation, in line 20 the sentence was modified.
Comment 2: Please review the reference by Dr. Pasic L et al. in J Fungi 2020(6):259 (doi:10.3390/jof6040259) and remove the sentence "This is the first report of P. jirovecii genotypes in patients with HIV in Chile".
According with the reviewer´s recommendation, the sentence "This is the first report of P. jirovecii genotypes in patients with HIV in Chile" was deleted.
Comment 3: Lines 423-424 should be also deleted.
Done
Comment 4: Stating that Genotypes 2 and 3 are the most prevalent in Temuco and are different from previous reports from other geographical locations shall be commented on.
According to the reviewer's suggestion, in the current lines 356-361 of the discussion, information regarding to other geographic locations where there is a predominance of a different genotype was added.
Manuscript:
Comment 5: Please quote ref #72 and Dr. Pasic in line 90. (after PCP cases).
In the current line 96 both references [32–33], were added.
Comment 6: Line 65: Reference 17 does not refer to nested PCR. Please remove nested PCR or change the reference to describe the molecular tools available to characterize Pneumocystis jirovecii isolates from immunocompromised patients. Another paper from Dr. Beard is suggested: "Strain Typing Methods and Molecular Epidemiology of Pneumocystis Pneumonia" in EID 2004.
A suggested by the reviewer in the current line 65, the reference “17” was replaced by Beard et al. 2004. doi: 10.3201/eid1010.030981
Comment 7: When describing the history of TMP/SMZ as a treatment of PCP please quote the original manuscript by Hughes WT in Can Med Assoc Journal 1975.
In the current line 72 the reference indicated by the reviewer was added to the manuscript.
Comment 8: Line 205
“Reads: ...patients diagnosed with HIV were included....
Should read: ... diagnosed with HIV and lower respiratory tract infection were included....
In the current lines 214-215 the sentence was corrected.”
Comment 9: Were the patients consecutive?
No, the patients were not consecutive, they were diagnosed with PCP completely at random.
Comment 10: Is the etiology of respiratory infection in non-PCP cases available? Was the etiology determined in a proportion of the cases? Which etiologies?
Because the focus of the study was on PCP patients, we focused exclusively on the characterization of these cases; we do not have information regarding the etiology of respiratory symptoms in non-PCP HIV patients. Although we could obtain it from the review of the patients' clinical records.
Discussion:
Comment 11: Line 320: Reference 49 refers to antiretroviral therapy. Please add a reference to indicate that prophylactic therapy with SMZ-TMP started in 1989. The description of TMP SMZ prophylaxis was published in 1975.
As recommended by the reviewer in the current line 333 the reference Fox et al. 1990 was added (reference N° 50 - A prospective, randomized, double-blind study of trimethoprim-sulfamethoxazole for prophylaxis of infection in renal transplan-tation: clinical efficacy, absorption of trimethoprim-sulfamethoxazole, effects on the microflora, and the cost-benefit of prophylax-is. Am J Med. 1990, 89, 255–274).
Comment 12: “Line 343 - 346 Single-round PCR has an excellent correlation with microscopy in HIV-infected patients please comment on this and provide a reference”.
As recommended by the reviewer, in the current line 348-351 we added information about the correlation between the detection of P. jirovecii with the addition of the following reference:
Franconi, I.; Leonildi, A.; Erra, G.; et al. Comparison of different microbiological procedures for the diagnosis of Pneumocystis jirovecii pneumonia on bronchoalveolar-lavage fluid. BMC Microbiol 2022, 22, 143. https://doi.org/10.1186/s12866-022-02559-1
Comment 13: “Line 352 The most prevalent mtLSU genotype in isolates from Santiago in Chile was genotype 4. (line 352) (Dr. Pasic et. al).”
In the current lines 356-357 the phrase indicated by the reviewer was added to the discussion.
Comment 14: The manuscript would benefit from a review of English grammar. For example:
“Line 40: Reads: ..., which has emerged as one of the main... "has" should be deleted as this happened in the 1980s or 1990s.”
The correction was made.
“line 46 Reads: dis-semination... The hyphen should be deleted. Same as in line 163.”
The correction was made in the current line 46 and 174 respectively.

Reviewer 2 Report
Dear authors, thank you for this interesting paper about P. jirovecii infection in Chile.
However, the article should be shortened and more concise in all sections.
In the introduction, you do not mention qPCR (L60) which is now the recommended molecular method for PCP diagnosis.
There is nothing about the DHFR gene and the mutations that could have a consequence in sulfonamide susceptibility, it would be interesting to add a little paragraph about this in the introduction.
Use DHPS after the first occurrence in entire in the text (L72).
Please write P. jirovecii in italic throughout the manuscript.
Sentence L81-88 should be shortened, just keep the gene sequencing.
M&M: "53 adult patients" (L97) is a result. L123: pAZ102 instead of pAZI02. 2.7 should be moved in 2.1.
Results
3.1. is redundant with Tab1. Please shorten this paragraph
3.2. Please specify which sample type was tested positive only with PCR (sputum?). Maybe you could add this information in a supplementary column “positive diagnosis” in Tab.2.
In Tab 2, how patient 3 could have received previously SMX-TMP as it was a new HIV-diagnosis? For another indication?
Maybe you could add the DHFS mutations in Tab 2 and thus delete Tab 4.
Fig 1 and 2 are not essential.
L243: “oral or IV” instead of “oral and IV”… L250: “or to the critical” instead of “and to the critical”
L251: on which day after diagnosis was assessed the evolution?
Tab 3 and Fig 3 are redundant: maybe keep only Tab 3.
L283: “ch”?
Delete sentence “Significant…” L301-302. Add “Only” at the beginning of next sentence.
L311-313: mtSU: OR387295 to OR387308;
L313-315: DHPS: OR398332 to OR398344;
Discussion has to be shortened.
Ref 50 does not seem very valid (P. jerovesi?), please delete it.
L345-346: delete “of the mitochondrial genome”
L351: genomes 2 and 3 are not really predominant: 33% and 26% vs 20% for genotype 3, is it a significant difference?
L353: delete “that is, those with more than one genotype”
L363-369 is not essential.
L382: microorganisms in italics please
Discussion about the prevalence of DHPS mutations in different countries: instead of %age, give raw numbers when small number of isolates are studied (ref 70, 71 for example). Ref 71 is in Uganda not in Mozambique. Nothing about the prevalence of DHPS mutations in Europe, why? Maybe you should focus on the American continent only… because the discussion is not geographically exhaustive.
The conclusion should be shortened, just focus on your results. You can not say that genotypes 2 and 3 are more prevalent (small sample of cases, and probably no significant differences)
Abstract: instead of DHPS mutation prevalence of 6.7%, write 1/15.
The English language needs some corrections I haven't done
Author Response
Reviewer 2.
Comment 1: Dear authors, thank you for this interesting paper about P. jirovecii infection in Chile. However, the article should be shortened and more concise in all sections.
As recommended by the reviewer some sections of the manuscript were reduced.
Comment 2: “In the introduction, you do not mention qPCR (L60) which is now the recommended molecular method for PCP diagnosis”.
According to the recommendation of the reviewer, in the current line 65-68, information about qPCR was added.
Comment 3: “There is nothing about the DHFR gene and the mutations that could have a consequence in sulfonamide susceptibility, it would be interesting to add a little paragraph about this in the introduction”.
According to the recommendation of the reviewer, in the current line 77 and 87, information about the DHFR gene and the mutations was added.
Comment 4: “Use DHPS after the first occurrence in entire in the text (L72)”.
In the current line 80-81 the full name of the gene encoding the enzyme dihydropteroate synthase was added, however, in the full text, to facilitate better understanding of readers, we prefer to mention the enzyme with the name without abbreviation, that is, "dihydropteroate synthase" and the encoding gene as "DHPS".
Comment 5: “Please write P. jirovecii in italic throughout the manuscript”.
The entire text was revised and all scientific names were revised and written in italics.
Comment 6: “Sentence L81-88 should be shortened, just keep the gene sequencing”.
According to the reviewer's recommendations, in the current line 90-94 the information referring to other molecular techniques was eliminated and only that related to gene sequencing was maintained.
M&M:
Comment 7: "53 adult patients" (L97) is a result.
In accordance with the reviewer's recommendations, In the L97 (current line 103), information regarding the number of patients included in the study was eliminated.
Comment 8: “L123: pAZ102 instead of pAZI02”.
In the L123 (current line 134) the primers name was corrected.
Comment 9: “2.7 should be moved in 2.1”.
According to the reviewer's recommendations, the information regarding obtaining the demographic data of the patients included in the study was transferred to item 2.1.
Results
Comment 10: 3.1. is redundant with Tab1. Please shorten this paragraph
Following the reviewer's recommendation, the section of results “3.1. Patients studied, clinical and demographic information” was shortened by removing repeated information from the table 1.
Comment 11: 3.2. Please specify which sample type was tested positive only with PCR (sputum?). Maybe you could add this information in a supplementary column “positive diagnosis” in Tab.2.
Following the reviewer's recommendation. In lines 233-234 the samples positive only by PCR were added.
Comment 12: “In Tab 2, how patient 3 could have received previously SMX-TMP as it was a new HIV-diagnosis? For another indication?”
The patient 3 had been referred from another hospital and was already receiving SMX-TMP in the 10 days prior to sputum sampling, due to the high suspicion of PCP.
Comment 13: “Maybe you could add the DHFS mutations in Tab 2 and thus delete Tab 4.”
We prefer to keep table 4 since it is much clearer for readers about the 4 mutations described in the DHPS gene along with the amino acid changes generated and the percentage of each one found in the study.
Comment 14: “Fig 1 and 2 are not essential”.
We consider it important to be able to show readers, through the images, the findings found in the lung images of the patients as well as the abundant number of cystic forms present in their respiratory samples, which reinforces the fact that they are indeed patients presenting with P. jirovecii pneumonia and not colonization.
Comment 15: “L243: “oral or IV” instead of “oral and IV”… L250: “or to the critical” instead of “and to the critical”
According to the reviewer's recommendation, in the current lines 252 and 258 respectively, the changes were made.
Comment 16: L251: on which day after diagnosis was assessed the evolution?
The patient's evolution was evaluated within 15 to 30 days after the end of treatment.
Comment 17: “Tab 3 and Fig 3 are redundant: maybe keep only Tab 3”.
We prefer to maintain both the phylogenetic tree and the table since in this way all the information regarding the different genotypes found, the nucleotide changes, prevalence percentages of each one, as well as the phylogenetic relationships between genotypes that can be displayed in figure 3.
Comment 18: “L283: “ch”?”
In the current line 290, the phrase was modified in order to clarify the information provided.
Comment 19: “Delete sentence “Significant…” L301-302. Add “Only” at the beginning of next sentence”.
In the current line 318 the sentence indicated by the reviewer was deleted and “only” was added at the beginning of next sentence.
Comment 20: L311-313: mtSU: OR387295 to OR387308; L313-315: DHPS: OR398332 to OR398344.
For both cases (current line 326-328), the change suggested by the reviewer was incorporated.
Discussion has to be shortened.
According to the observation made by the reviewer, the discussion was shortened.
Comment 21: Ref 50 does not seem very valid (P. jerovesi?), please delete it.
According to the reviewer's recommendation, the information related to reference N°. 50 along with said reference were eliminated from the article.
Comment 22: L345-346: delete “of the mitochondrial genome”
In the current line 354 the phrase “of the mitochondrial genome” was deleted.
Comment 23: L351: genomes 2 and 3 are not really predominant: 33% and 26% vs 20% for genotype 3, is it a significant difference?
In the current lines 356-360, although the difference is not very marked, genotypes 2 and 3 were the most frequent in the group of patients studied. Likewise, according to the observation made by the reviewer, the phrase was modified so that it is evident that the frequency of said genotypes was slightly higher compared to genotype 1.
Comment 24: L353: delete “that is, those with more than one genotype”
In the current line 362 the phrase “that is, those with more than one genotype” was deleted.
Comment 25: L363-369 is not essential.
In current line 347 ​​and in accordance with the recommendation, the paragraph indicated by the reviewer was removed.
Comment 26: “L382: microorganisms in italics please”
The uploaded file had all the scientific names in italics, but for some reason they now appear without italics. In any case, all scientific names were again reviewed and corrected.
Comment 27: Discussion about the prevalence of DHPS mutations in different countries: instead of %age, give raw numbers when small number of isolates are studied (ref 70, 71 for example).
In accordance with the reviewer's recommendations, the detail of the number of samples referring to each of the mentioned percentages was added.
Comment 28: “Ref 71 is in Uganda not in Mozambique”.
The country in the current reference 72 (line 394) was modified.
Comment 29: Nothing about the prevalence of DHPS mutations in Europe, why? Maybe you should focus on the American continent only… because the discussion is not geographically exhaustive.
According to the observation made by the reviewer, in lines 394-395 information regarding mutations in the DHPS gene carried out in Europe was added to the discussion of the article.
Comment 30: The conclusion should be shortened, just focus on your results. You can not say that genotypes 2 and 3 are more prevalent (small sample of cases, and probably no significant differences)
The conclusion was shortened just focus on our results. Information about the most prevalent genotypes was deleted.
Comment 31: Abstract: instead of DHPS mutation prevalence of 6.7%, write 1/15.
In accordance with the reviewer's recommendations, the number of mutated strains was added to the abstract.

Reviewer 3 Report
1. Line 24, please define the abbreviation DHPS
2. For the in-silico binding affinity analysis
1) It’s better to include other controls, such as another irrelevant drug binding affinity with the mutated DHPS, since it could be a false positive result that Thr55Ala-Glu98Gln model showed a lower binding affinity. Thr55Ala-Glu98Gln model could display lower affinity to other irrelevant drugs.
2) Does the Thr55Ala-Glu98Gln model show a higher binding affinity with pABA?
3) The binding affinity result of 57/96/98 and 55/96/98 combinations are missing.
3. Better to elucidate more on the relationship between antibiotic binding affinity and antibiotic resistance in the discussion.
4. Line 412-415, there are a few papers suggesting the single mutation in codon 55, 57, 96 and 98 are associated with TMZ-SMX treatment failure and worse clinical outcomes. Please address what makes your analysis different from the existing paper, otherwise, it's too early to draw a conclusion that “therefore, on their own are unlikely to be responsible for resistance development”, simply based on the in-silico evidence.
Author Response
Reviewer 3.
Comment 1: Line 24, please define the abbreviation DHPS
The definition of the DHPS abbreviation was added to the abstract
For the in-silico binding affinity analysis
Comment 2: It’s better to include other controls, such as another irrelevant drug binding affinity with the mutated DHPS, since it could be a false positive result that Thr55Ala-Glu98Gln model showed a lower binding affinity. Thr55Ala-Glu98Gln model could display lower affinity to other irrelevant drugs.
According with the reviewer´s reccomendation, as part of our control experiments, we conducted docking analyses involving the wild type enzyme model with sulfamethoxazole, dihydropterin, pABA, and DMSO. This analysis is presented as supplementary material (S1). We observed significant differences (P < 0.05) in binding affinities, with sulfamethoxazole exhibiting higher affinity compared to the other molecules. Particularly, dihydropterin and pABA are substrates of dihydropteroate synthase, while DMSO serves as a negative control, which lacks antifungal activity on its own. The observed binding affinity of sulfamethoxazole in these experiments provides a relevant benchmark for evaluating the binding affinities of sulfamethoxazole with the mutant dihydropteroate synthase models.
This information was added in “3.5 In silico analysis results”.
Comment 3: Does the Thr55Ala-Glu98Gln model show a higher binding affinity with pABA?
Although the analysis with pABA does not respond to the objective of the article, the control, which could be interpreted as the normality of the system, indicates that pABA has a lower binding affinity than sulfamethoxazole. This trend is probably maintained in the models with combinations of mutations, as can be seen with sulfamethoxazole, which do not really have large variations, that is, it is unlikely that the binding affinity of pABA is above sulfamethoxazole in the Thr55Ala-Glu98Gln model.
Comment 4: The binding affinity result of 57/96/98 and 55/96/98 combinations are missing.
The binding affinity of 57/96/98 and 55/96/98 combinations were included in the new figure 4
Comment 5: Better to elucidate more on the relationship between antibiotic binding affinity and antibiotic resistance in the discussion.
In the current line 374-378 information about the relationship between antibiotic binding affinity and antibiotic resistance with their respective references was added.
Comment 6: Line 412-415, there are a few papers suggesting the single mutation in codon 55, 57, 96 and 98 are associated with TMZ-SMX treatment failure and worse clinical outcomes. Please address what makes your analysis different from the existing paper, otherwise, it's too early to draw a conclusion that “therefore, on their own are unlikely to be responsible for resistance development”, simply based on the in-silico evidence.
Our in silico analysis differs from others previously carried out in that it was performed with enzymatic models obtained from the DHPS sequences of P. jirovecii, without using models from other microorganisms that until now had been used to predict the behavior of P. jirovecii. However, we agree with the reviewer that it is too hasty to ensure that this would be the in vivo behavior that P. jirovecii would present in the presence of said mutations. That is why in the final part of the discussion we insist that future supported studies are necessary (with clinical information of patients) that allow validity and/or confirm our findings.

Round 2
Reviewer 1 Report
The current version is much improved and easier to read.
Two minor issues:
Line 310 should read PABA
The reference for Figure 4 A is missing.
Thank you to the authors for considering our suggestions.
English is generally adequate.
Author Response
Reviewer 1.
The current version is much improved and easier to read.
Two minor issues:
Comment 1: “Line 310 should read PABA”
As suggested by the reviewer “pABA” was replaced by “PABA” on line 310 and throughout the manuscript.
Comment 2: “The reference for Figure 4 A is missing”.
The reference for figure 4.A is indicated in line 302.
Comment 3: “Thank you to the authors for considering our suggestions”.
We thank the reviewer for their comments and suggestions on the manuscript. Without any doubt, they have allowed to improving the article.

Reviewer 3 Report
The authors have addressed my previous comments, and the manuscript can be accepted in present form.
Author Response
We thank the reviewer for their comments and suggestions on the manuscript. Without any doubt, they have allowed to improving the article.